# Progress in Precision Medicine for Head and Neck Cancer

**DOI:** 10.3390/cancers16213716

**Published:** 2024-11-04

**Authors:** Sanaz Vakili, Amir Barzegar Behrooz, Rachel Whichelo, Alexandra Fernandes, Abdul-Hamid Emwas, Mariusz Jaremko, Jarosław Markowski, Marek J. Los, Saeid Ghavami, Rui Vitorino

**Affiliations:** 1Department of Human Anatomy and Cell Science, University of Manitoba College of Medicine, Winnipeg, MB R3E 0J9, Canada; s.vakili11@gmail.com (S.V.); am.barzegar.behrooz@gmail.com (A.B.B.); rwhichel@uoguelph.ca (R.W.); 2Department of Medical Sciences, Institute of Biomedicine—iBiMED, University of Aveiro, 3810-193 Aveiro, Portugal; 3Guelph College of Biological Science, University of Guelph, Guelph, ON N1G 2W1, Canada; alexandra.gil.fernandes@ua.pt; 4Core Lab of NMR, King Abdullah University of Science and Technology (KAUST), Thuwal, Makkah 23955-6900, Saudi Arabia; abdelhamid.emwas@kaust.edu.sa; 5Division of Biological and Environmental Sciences and Engineering (BESE), King Abdullah University of Science and Technology (KAUST), Thuwal, Makkah 23955-6900, Saudi Arabia; mariusz.jaremko@kaust.edu.sa; 6Department of Laryngology, Faculty of Medical Sciences in Katowice, Medical University of Silesia, 40-027 Katowice, Poland; jmarkow1@poczta.onet.pl; 7Biotechnology Center, Silesian University of Technology, 44-100 Gliwice, Poland; mjelos@gmail.com; 8Academy of Silesia, Faculty of Medicine, Rolna 43, 40-555 Katowice, Poland; 9Paul Albrechtsen Research Institute, Cancer Care Manitoba, Winnipeg, MB R3E 0V9, Canada; 10Department of Human Anatomy and Cell Science, Max Rady College of Medicine, Rady Faculty of Health Sciences, University of Manitoba, Winnipeg, MB R3E 0V9, Canada; 11LAQV/REQUIMTE, Department of Chemistry, University of Aveiro, 3810-193 Aveiro, Portugal; 12UnIC@RISE, Department of Surgery and Physiology, Faculty of Medicine, University of Porto, 4099-002 Porto, Portugal

**Keywords:** precision medicine, biomarkers, genetic alterations, head and neck cancer, targeted therapy

## Abstract

Head and neck squamous cell carcinoma (HNSCC) is a deadly form of cancer, affecting areas like the mouth, throat, and larynx. This review explores the complex molecular pathways involved in HNSCC development and progression, focusing on the role of microRNAs (miRNAs)—small molecules that regulate gene expression. We aim to provide a comprehensive overview of how miRNAs influence HNSCC, their potential as diagnostic and prognostic markers, and their use in developing new targeted therapies. We also discuss promising nanotechnology-based approaches for delivering miRNA therapies more effectively. By synthesizing the current knowledge on miRNAs in HNSCC, this research may help identify new biomarkers for early detection and prognosis, as well as novel therapeutic targets. Ultimately, these insights could lead to improved personalized treatments and better outcomes for HNSCC patients.

## 1. Introduction

Head and neck cancer (HNC) is the seventh most common cancer worldwide and represents a diverse group of malignancies. Its heterogeneity results from different histologic subtypes, etiologic factors, and molecular features. This diversity is influenced by factors such as viral infections, environmental influences (e.g., tobacco and alcohol consumption), and various genetic mutations. Understanding these differences is crucial for accurate diagnosis, prognosis, and the development of targeted therapeutic strategies [1]. Virus-related tumours play an important role in the development of HNC. In particular, Epstein–Barr virus (EBV) is closely associated with nasopharyngeal carcinoma, especially in subtypes 2 and 3. EBV-positive nasopharyngeal carcinomas have distinct molecular and clinical features compared to EBV-negative tumours, which contribute to their unique behaviour and response to treatment. In addition, human papillomavirus (HPV) is associated with oropharyngeal carcinomas, further emphasizing the importance of viral etiology in HNC. This cancer’s diverse molecular basis and clinical manifestations necessitate a comprehensive understanding at both the molecular and clinical levels to help identify unique and shared pathways, allowing for improved diagnosis and clinical management through the development of targeted therapies. Through the examination of this cancer, potential innovations in treatment strategies, advanced personalized medicine, and improved patient outcomes can be emphasized [2,3].

The global burden of HNC is significant due to its profound impact on mortality and morbidity as essential functions such as breathing, eating, and speaking are impaired. However, the prevalence and outcomes of HNC vary significantly from region to region, influenced by differences in exposure, risk factors, and access to healthcare services [4,5].

HNC is significantly influenced by genetic predispositions and environmental factors, with risk factors such as age and family history [6,7,8]. Susceptibility to head and neck cancer increases with age, and a family history may suggest a genetic predisposition that heightens the risk [9]. With an aging global population, the incidence and mortality rates of this cancer are expected to rise. Symptoms commonly appear in advanced stages, complicating optimal cancer treatment and significantly reducing survival rates. This highlights the need for regular screening, heightened awareness, especially for high-risk groups, and the discovery of novel biomarkers to improve both treatment strategies and cure rates. Addressing these challenges requires comprehensive strategies, including prevention, early detection, and tailored treatments [3,10] (Figure 1).

### 1.1. The Importance of Molecular Studies in Understanding Head and Neck Cancer

Molecular studies have revolutionized our understanding of HNC, revealing the intricate genetic and molecular landscapes underlying this disease. In HNC, molecular research has shown the central role of viral aetiologies, particularly human papillomavirus (HPV) and Epstein–Barr virus (EBV), in the pathogenesis of these cancers. HPV-positive oropharyngeal cancers, for example, are now recognized as distinct cancers with better prognoses and responses to treatment compared to HPV-negative cancers. The identification of these viral markers has facilitated the development of targeted therapies and preventive vaccines, such as the HPV vaccine, which have the potential to significantly reduce the incidence of these cancers [11,12].

Molecular profiling of HNC has identified key genetic alterations, including mutations in TP53, NOTCH1, and PIK3CA and amplifications in oncogenes like epidermal growth factor receptor (EGFR). These findings have led to targeted therapies, such as EGFR inhibitors, tailored to each patient’s tumour. Understanding the HNC molecular pathways has improved patient stratification for personalized treatment, enhancing outcomes and reducing toxicity [13,14]. In HNC, it is important to distinguish between genetic alterations and molecular features. Genetic alterations refer to mutations in key genes such as TP53, NOTCH1 and PIK3CA, which directly affect disease progression by impairing basic cellular processes such as DNA repair, cell cycle regulation and apoptosis. On the other hand, there are molecular features that affect signalling pathways, such as the EGFR and PI3K/AKT pathways, which control cellular proliferation, survival, and metastasis. These pathways are often influenced by both genetic mutations and environmental factors, and their abnormal activation plays a critical role in HNC tumour behaviour. By distinguishing between these genetic and molecular elements, we can better understand how these different factors contribute to the complexity of HNC biology and progression. The insights gained from molecular studies have not only deepened our understanding of HNC biology but have also revolutionized the clinical treatment of this condition. By identifying critical genetic and molecular alterations, researchers and clinicians can improve diagnostic accuracy and personalize medicine plans based on more effective, targeted therapies, thus improving treatment outcomes. Among the essential regulators of the molecular profile in head and neck cancer, microRNAs have emerged as a pivotal component, influencing gene expression and contributing to the complexity of tumour biology. 

### 1.2. Introduction to miRNAs in Head and Neck Cancer

MicroRNAs (miRNAs) are small (18–25 nucleotides), non-coding RNA molecules that regulate gene expression post-transcriptionally. These molecules play a crucial role in numerous biological processes such as cell proliferation, apoptosis, metastasis, and response to therapy. MicroRNAs can act as oncogenes or tumour suppressors, making them invaluable tools for understanding cancer biology. Their potential as biomarkers for early diagnosis and prognosis, and as therapeutic targets underlines their importance for precision medicine.

In head and neck cancer (HNC), miRNAs have been shown to be important regulators that influence the molecular landscape of the disease. They serve as potential biomarkers for diagnosis, prognosis, and therapeutic intervention. Despite advances in treatment, HNC patients are often diagnosed at late stages, have a high relapse rate, and are resistant to conventional therapies. For this reason, miRNAs are of great importance in exploring therapeutic strategies to improve patient outcomes. This section provides a detailed overview of miRNAs in HNC and their role in the development of personalized medicine.

### 1.3. Molecular Insights into Head and Neck Cancer

The diagnosis of head and neck cancer (HNC) relies primarily on physical examinations, as there are no efficient and standardized screening methods. This often leads to late detection, when the cancer is more advanced and more difficult to treat. Such diagnostic limitations have led to a growing interest in molecular biomarkers to improve accuracy and finding less invasive diagnostic options [14,15], Understanding the molecular subtleties of HNC is critical for developing personalized treatment strategies and improving treatment outcomes.

Major risk factors for HNC include tobacco use, alcohol consumption, and oncogenic viruses such as HPV and EBV [16,17,18]. Molecular profiling has uncovered several genetic alterations, including mutations in tumour suppressor genes such as TP53 and PIK3CA and amplifications in oncogenes such as EGFR [11,12]. These findings have paved the way for targeted therapies to improve patient stratification and treatment outcomes. The identification of viral aetiologies and genetic alterations has also underscored the potential of preventive measures such as HPV vaccination to reduce the incidence of HNC [19]. This section highlights the global impact and molecular basis of HNC to provide a clear understanding of the evolution of these cancers and the promising avenues for targeted treatment approaches (Table 1).

## 2. Signature Proteins/Genes in Head and Neck Cancer

### 2.1. Literature-Driven Insights

The importance of identifying the pivotal proteins and genes involved in this malignancy has become increasingly apparent in order to acquire a greater understanding of HNC. In HNC, human papillomavirus (HPV) oncoproteins E6 and E7 play a crucial role, particularly in oropharyngeal carcinoma, by inactivating the tumour suppressor proteins p53 and retinoblastoma (Rb), respectively [11,12]. In addition, mutations in TP53, CDKN2A, and PIK3CA and amplifications in EGFR are common in HNC and drive tumourigenesis via different pathways.

Several other proteins also play a crucial role in the progression of HNC. Tissue inhibitors of metalloproteinases (TIMPs) are significantly involved in the regulation of extracellular matrix (ECM) remodelling and metastasis through their inhibitory effect on matrix metalloproteinases (MMPs) and thus play a crucial role in the tumour microenvironment [28]. Similarly, galectin-3 (Gal-3) is involved in various aspects of cancer progression, including proliferation, angiogenesis, metastasis, and immune evasion [29].

In head and neck cancer, MMP25 expression correlates with prognosis and the degree of immune infiltration, particularly for CD4+ memory T cells, and is involved in oncogenic signalling pathways such as p53 signalling and PI3K/AKT/mTOR signalling. This suggests its crucial role in cancer progression and immune responses [30]. Fibronectin 1 (FN1) is significantly overexpressed in head and neck squamous cell carcinoma (HNSCC) and correlates with higher pathologic stages and a poor prognosis. Its downregulation suppresses the proliferation, migration, and invasion of HNSCC cells and inhibits M2 polarization of macrophages in vitro, suggesting its role in promoting HNSCC progression and immune modulation [31]. This indicates the role of fibronectin in modulating the post-transcriptional changes that lead to cancer progression [32].

TIMPs, Gal-3, MMPs, and fibronectin are integral to the dynamics of the tumour microenvironment in both head and neck cancer, influencing tumour growth, metastasis, and immune responses. These molecules can potentially serve as biomarkers for diagnosis and prognosis, and as targets for therapeutic intervention. Understanding their role and interactions is essential for developing personalized treatment approaches and improving treatment outcomes for this cancer (Table 2).

### 2.2. Database-Driven Insights

To identify datasets for HNC, a dataset search was performed in the Genome Expression Omnibus (GEO). These datasets contain gene expression data for HNC (GSE227919, GSE142083, GSE138206, and GSE130605). The most significant differentially expressed genes (DEGs) were identified for HNC. Further analysis identified 37 potential biomarkers for HNC, of which, the top 5 DEGs were highlighted. In HNC, the most upregulated genes are MMP1 (matrix metallopeptidase 1), SERPINE1 (serpin family E member 1), PLAU (urokinase-type plasminogen activator), AURKA (Aurora kinase A), and HMGA2 (high-mobility group AT-hook 2). Therefore, these genes may serve as biomarkers to distinguish patients with HNC from those without this cancer. This conclusion is supported by literature data from previous studies (Table 3).

In HNC, the molecular profile is dominated by critical proteins and genes that drive tumourigenesis, impact patient prognosis, and serve as targets for therapy. AURKA, a member of the Aurora serine/threonine kinase family, plays a role in maintaining chromosome stability and is expressed in many tissues. It is overexpressed in various tumours, including glioblastoma, cervical cancer, and HNC [42,43]. AURKA has been identified as a novel therapeutic target and predictor of overall survival (OS) in HNC [44]. High expression levels of AURKA are associated with the progression of HNC and poor prognosis, especially in HPV-negative tumours. However, AURKA is also known to be upregulated by HPV E7, which increases tumour cell proliferation. Overexpression of AURKA inhibits apoptosis by phosphorylating p73, a protein related to programmed cell death [43]. HMGA2, which belongs to the high-mobility group AT-hook (HMGA) gene family, encodes a protein that regulates gene expression, cell replication, and repair. HMGA2 has been identified as an oncogene associated with cancer development when expressed at high levels [45]. Overexpression of HMGA2 is associated with processes related to the progression of HNC, such as significant invasiveness, metastasis, epithelial-mesenchymal transition, dysregulated proliferation, and migration of tumour cells [45,46]. The expression of HMGA2 influences OS in HNC, with a high expression indicating a poor prognosis [46,47]. MMP1, a member of the matrix metalloproteinase family, is involved in extracellular matrix (ECM) degradation by cleaving collagen IV and gelatine [48]. MMP1 is overexpressed in HNC, and its expression correlates with the T stage (TNM classification) and cancer progression through promoting tumour invasion [49]. PLAU, which belongs to the S1 serine peptidase of the plasminogen activator family, encodes a secreted serine protease that converts plasminogen to plasmin [49,50]. This enzyme promotes ECM degradation and facilitates tumour cell invasion and metastasis. In HNC samples, PLAU expression is upregulated and is associated with lower overall survival, poor disease-free survival, and an increased risk of developing HNC [49]. Overexpression of PLAU also leads to upregulation of MMP1. Suppression of PLAU inhibits HNC cells’ proliferation, migration, and invasion capacities, while its overexpression has the opposite effect [49]. SERPINE1, a serine protease inhibitor family member, is overexpressed in HNC samples [51,52]. Overexpression of SERPINE1 is associated with the N stage (TNM classification) and may play a role in tumour angiogenesis and metastasis. Its inhibition is associated with reduced cell proliferation and invasion [53]. SERPINE1 is also a potential prognostic biomarker, as its expression correlates with disease-free survival (DFS) in HNC [52]. Overexpression of SERPINE1 promotes resistance to radiotherapy in HNC patients [51]. These essential proteins and genes play crucial roles in the development and progression of HNC and serve as potential biomarkers for diagnosis and prognosis, and as targets for therapeutic intervention. Thus, understanding their functions and interactions is essential for developing personalized treatment approaches and improving treatment outcomes in this cancer.

## 3. Molecular Pathways Involved in Head and Neck Cancer

As with the genetic landscape, the molecular profiles of HNC have unique features. The dysregulation of signalling pathways such as the EGFR, PI3K/AKT, and RAS-RAF-MEK pathways is critical in HNC, affecting cell cycle regulation, cellular metabolism, and apoptosis. In addition, in oropharyngeal carcinoma, HPV adds a unique viral mechanism to tumour development, mainly through the viral oncoproteins E6 and E7, which affect tumour suppressor pathways such as p53 and Rb [14,54].

Functional enrichment analysis was performed on the string platform using the DEGs identified for HNC (n = 37). This analysis aimed to assess the functions of these genes and their association with HNC. The DEGs identified for HNC were associated with the cancer and several signalling pathways and molecular functions related to cancer development and progression, including ERBB and TGF-β signalling, ECM degradation, and focal adhesion. In addition, these genes have been linked to other diseases, such as gastrointestinal and pancreatic cancers and cardiovascular disorders, where they have been found expressed in tissues of the cardiovascular system. ERBB signalling includes elements of the HER family: HER1 (EGFR), HER2 (ErbB2), HER3 (ErbB3), and HER4 (ErbB4) [55,56,57]. EGFR plays an important role in signalling pathways critical to cancer progression, including Ras/Raf/MEK/ERK and PI3K/AKT/mTOR. In HNC, studies have indicated that EGFR mutations significantly impact prognosis more than TP53 mutations [57,58]. Other HER family members are also involved in developing HNC and show upregulated expression in patients [57,59]. TGF-β signalling regulates epithelial-to-mesenchymal transition as well as cell differentiation and proliferation in HNC. It plays a dual role by inhibiting dysregulated cell proliferation in the early stages while promoting invasion and metastasis at later stages [60].

The ECM is essential for physical interactions with cancer cells and serves as a reservoir for various ligands and growth factors that influence the tumour microenvironment. During cancer progression, focal adhesion is lost, which decreases cellular adhesion and allows cancer cells to permeate the ECM, altering its composition and organization, ultimately leading to degradation. This process is promoted by increased MMP expression in cancer [61]. After degradation of the ECM, the cancer cells enter the bloodstream, invade other tissues and metastasize [62].

As mentioned, HNC has alterations in the PI3K/AKT/mTOR signalling pathway, highlighting its universal role in cancer progression. Targeted therapies in cancers aim to inhibit this pathway to slow tumour growth and reduce tumour survival [63]. In HNC, targeted agents against EGFR, such as cetuximab, have shown clinical benefits by exploiting the abnormal activation of growth factor signalling pathways [4].

Unique to HNC is the influence of viral oncogenes in HPV-associated cancers, necessitating therapeutic strategies that target viral proteins or their effects on cellular mechanisms. Immunotherapy that harnesses the immune system’s ability to recognize and destroy cancer cells has shown promise in HNC, particularly PD-1/PD-L1 inhibitors such as pembrolizumab and nivolumab [64]. The distinct molecular pathology of HNC emphasizes the need for cancer-specific targeted therapies that reflect their unique molecular basis [65]. Understanding these pathways helps in the development of targeted therapies and the identification of potential biomarkers for better disease management [64,65].

## 4. Biomarkers in Head and Neck Cancer: Progress and Challenge

Biomarkers are crucial for offering insights into the molecular basis of the disease and providing valuable diagnostic and prognostic information for HNC. Notable biomarkers include DNA methylation patterns, circulating microRNAs, and proteins such as EGFR and HPV oncoproteins (E6 and E7). These markers help detect and assess the cancer’s aggressiveness at an early stage, enabling customized therapeutic measures [66,67,68]. Extensive studies have indicated the significant clinical potential of these biomarkers in HNC. Emerging biomarkers, including gene expression profiling in HNC, represent the next frontier in cancer diagnostics [69,70]. These novel biomarkers promise to improve the accuracy of cancer detection, prognosis, and monitoring, and additionally has the potential to revolutionize personalized medicine in oncology. The integration of biomarker testing into clinical practice is expected to significantly improve patient outcomes by enabling earlier detection, more accurate disease staging, and the development of optimal personalized treatment strategies based on individual tumour characteristics. For example, Filippini et al. (2023) [71] presented a systematic review of the current data on diagnostic and prognostic molecular biomarkers for head and neck cancer of unknown primary (HNCUP). Using an electronic database and following the PRISMA (Preferred Reporting Items for Systematic Reviews and Meta-Analyses) guidelines, they identified 704 articles, of which, 23 studies were carefully selected for inclusion. Fourteen studies focused on diagnostic biomarkers for HNCUP, with an emphasis on HPV and EBV, as these are significantly associated with oropharyngeal and nasopharyngeal cancers, respectively [71].

The review emphasized the prognostic importance of HPV, which is associated with improved disease-free treatment outcomes and overall survival rates. Currently, HPV and EBV are the only biomarkers for HNCUP used in the clinical setting. Improving molecular characterization and developing accurate tissue of origin classifiers are critical to advancing the diagnosis, accurate staging, and treatment of HNCUP patients [71]. The ongoing development and validation of new biomarkers is imperative to improve the precision of cancer treatment, minimize overdiagnosis and overtreatment, and ultimately improve patient survival and quality of life.

Despite advances in biomarker research for HNC, significant challenges remain in their application, validation, and clinical translation. Biomarkers have undoubtedly improved diagnostics and enabled earlier detection of HNC, particularly through the use of circulating microRNAs, DNA methylation, and proteins such as EGFR and HPV oncoproteins. For example, HPV status is a well-established prognostic marker, with HPV-positive tumours often responding better to treatment than HPV-negative tumours. Similarly, EGFR expression is used to guide the use of EGFR inhibitors such as cetuximab. However, these advances also have their limitations. While HPV and EGFR have demonstrated clinical utility, many biomarkers lack validation for routine clinical use due to inconsistent data from different studies, small sample sizes and diverse patient populations. PIK3CA mutations, which activate the PI3K/AKT signalling pathway, are also important genetic alterations that frequently occur in HNC. These mutations contribute to tumour growth and resistance to therapy. Although PI3K inhibitors have been explored as potential treatments, clinical success has been limited and further validation of PIK3CA as a predictive marker is needed. Other genetic and molecular factors such as CDKN2A (p16), which serves as a surrogate marker for HPV positivity, and alterations in genes such as NOTCH1, cyclin D1, and FGFR1 have shown potential as prognostic markers. For example, p16 is frequently used in clinical settings to assess HPV-related oropharyngeal cancer, while NOTCH1 mutations have been associated with both tumour suppression and oncogenesis in various contexts.

In addition, there are still no predictive biomarkers, i.e., those that can reliably predict the response to treatment. In HNC, for example, there is no predictive biomarker for EGFR inhibitors such as cetuximab, leading to a trial-and-error approach to treatment. The heterogeneity of HNC exacerbates this problem. Different tumour substances and aetiologies (e.g., HPV-positive versus HPV-negative) mean that no single biomarker is universally applicable. The small sample sizes in studies also increase the risk of false-positive results, making clinical application more difficult.

New biomarkers, such as gene expression profiling and circulating tumour DNA (ctDNA), represent promising avenues for improving early detection, treatment monitoring, and personalized care. MicroRNAs (miRNAs) are also gaining attention, both as biomarkers and as potential therapeutic targets. However, these newer biomarkers still need to be extensively validated before they can be integrated into routine clinical practice.

Another challenge is that there are no standardized testing protocols for different studies. Different platforms and methods for detecting biomarkers has led to discrepancies in the results, which makes it difficult to generalize the results. In addition, the regulatory hurdles in validating new biomarkers are considerable and often delay their translation into clinical practice.

To overcome these hurdles, future research should focus on conducting large-scale, multi-centre studies to validate promising biomarkers in different populations. In addition, testing protocols need to be standardized to ensure consistency in biomarker measurements. The identification of robust predictive biomarkers that can serve as a basis for treatment decisions is an essential prerequisite for advancing personalized medicine in HNC. The use of artificial intelligence and machine learning to analyze complex biomarker data could also help uncover new patterns that improve patient stratification and treatment outcomes. Although biomarkers hold great promise for the future of HNC diagnosis and treatment, further efforts are needed to overcome the challenges related to validation, standardization, and clinical translation. The integration of validated biomarkers into clinical practice will play a critical role in improving personalized treatment approaches and patient outcomes and will advance the field of oncology (Table 4).

## 5. Functional Roles of miRNAs in Head and Neck Cancer

Many research studies have investigated miRNAs’ role in HNC. Several microRNAs (miRNAs) have been identified as key regulators of signalling pathways critical for the pathogenesis of head and neck cancer (HNC). These miRNAs play an important role in tumourigenesis, progression, and metastasis, and their dysregulation has been linked to patient prognosis. For example, miRNAs such as miR-21, miR-155, and miR-375 are known to contribute to oncogenic processes or tumour suppression. Their potential as biomarkers for early diagnosis, therapeutic intervention, and prognosis in HNC has been extensively researched. However, despite the promising results, there are notable limitations in the current state of the research. One of the major challenges is the inconsistency of results across studies, often due to small sample sizes, differences in study design, and the lack of validation in larger, independent cohorts. These problems have led to inconsistencies in the identification of miRNAs that can be universally used as reliable biomarkers. In addition, many studies report false-positive results where the observed miRNA effects are not reproducible in larger populations or in different clinical settings. The lack of standardized protocols for the detection and quantification of miRNAs further complicates the interpretation of the data. Differences in the techniques used (e.g., qPCR vs. next-generation sequencing) and the methods of sample collection and processing contribute to discrepancies in miRNA profiles. Therefore, although miRNAs have significant potential as prognostic and diagnostic tools, there is an urgent need for rigorous validation studies with larger, diverse patient populations to confirm their clinical utility. In the future, it will be crucial to address these limitations by standardizing miRNA analysis methods, increasing the sample sizes, and conducting multi-centre studies. Only then can miRNAs be reliably integrated into clinical practice as part of precision medicine for HNC. 

### 5.1. OncomiRs in Head and Neck Cancer

As indicated, oncomiRs contribute to cancer progress by targeting tumour suppressor genes. In HNC, several oncomiRs have been identified that are involved in tumourigenesis, metastasis, and resistance to therapy [75]. MiR-21 was the first and is the most studied oncomiR in HNC. It is frequently overexpressed in HNC [76,77]. MiR-21 targets tumour suppressor genes such as those in the PTEN/AKT, TPM1, and PDCD4 pathways in HNC [78,79]. Overexpression of miR-21 promotes cell proliferation, invasion, and resistance to apoptosis in HNC [75]. Another oncomiR involved in HNC is miR-155, which targets several genes involved in cell cycle regulation and apoptosis [80]. For example, suppressor of cytokine signalling 1 (SOCS1) is one of the targetable genes that negatively regulates the JAK/STAT pathway [81]. MiR-155 elevation downregulates SOCS1 expression, increasing cell growth and cell survival, making it a potential marker for aggressive HNC [82]. Targeting miR-155 expression and the overexpression of SOCS1 could have a potent antitumour effect on HNC [83,84]. MiR-31 is also upregulated in HNC and plays a significant role in cancer progression, specifically in oral squamous cell carcinoma (OSCC) through the regulation of gene expression, cell proliferation, the cell cycle, cell survival, signal transduction, and epithelial–mesenchymal transition. In OSCC, it targets genes such as FIH (factor-inhibiting HIF) and LATS2 (large tumour suppressor kinase 2), which are involved in hypoxia responses and the Hippo signalling pathway, respectively [85]. MiR-31 upregulation contributes to proliferation, invasion, and angiogenesis, leading to more aggressive tumour behaviour in HNC [86]. Another group of miRNAs involved in HNC oncogenesis is the miR-221/222 cluster. These miRNAs target the cyclin-dependent kinase inhibitor p27Kip1 and the pro-apoptotic protein BMF [87]. MiR-221/222 upregulation increases cell cycle progression, reduces apoptosis, and enhances resistance to chemotherapy [88,89]. Ultimately, miR-196a is also overexpressed in HNC and promotes tumour progression, cell proliferation, migration, and invasion, leading to poor prognosis. Oncogenic miR-196a directly targets annexin A1 (ANXA1), a gene involved in cell differentiation and apoptosis. ANXA1 protein and mRNA were found to be downregulated in HNC tissues. However, miR-196a’s functional contribution to HNC development has remained unclear [90]. OncomiRs could potentially serve as biomarkers for early detection, prognosis, and monitoring the treatment response in HNC patients [91]. OncomiRs can be targeted by antagomirs (anti-miRNAs) [92] or miRNA sponges to inhibit their oncogenic effects [93]. Clinical trials are exploring the efficacy of such approaches in various cancers, including HNC [94].

### 5.2. Tumour-Suppressive miRNAs in Head and Neck Cancer

As outlined earlier, tumour-suppressive miRNAs inhibit cancer development and progression by targeting oncogenes and other genes involved in cell proliferation, invasion, and survival. In HNC, several tumour-suppressive miRNAs have been identified, playing crucial roles in maintaining cellular homeostasis and preventing malignant transformation. Through an analysis of recent findings, the key tumour-suppressive miRNAs in HNC were documented. MiR-34a is a well-known tumour suppressor miRNA involved in various cancers, including HNC. It targets multiple oncogenes, such as BCL2, MET, and CDK6. MiR-34a induces apoptosis, inhibits cell proliferation, and suppresses metastasis. MiR-34a is considered a novel and highly sensitive biomarker for HNSCC diagnostic applications [75]. MiR-200c is a part of the miR-200 family, known for its role in epithelial–mesenchymal transition (EMT). It was demonstrated that miR-200c expression is significantly down-regulated in OSCC tissues compared with adjacent non-tumour tissues, suggesting that miR-200c might be a potential prognostic biomarker for OSCC [95]. It inhibits ZEB1 and ZEB2, transcription factors that promote EMT and metastasis, thereby reducing invasion and metastasis [96]. MiR-200c-3p is another member of the miR-200 family shown to have a significantly downregulated expression in HNC tissues. In HNC cells, ectopic miR-200c-3p expression inhibits viability, clonogenicity, migration, and invasion [97]. Additionally, miR-375 is highly downregulated in HNC and functions as a tumour suppressor. It targets genes involved in cell cycle regulation, such as insulin-like growth factor 1 receptor (IGF-1R). MiR-375 upregulation inhibits growth and induces cell cycle arrest in the G0/G1 phase. It enhances apoptosis and radiosensitivity in HNC [98]. Reduced levels of miR-375 are associated with aggressive tumour behaviour and poor clinical outcomes [99]. MiR-145 is another tumour-suppressive miRNA that is significantly downregulated in HNC. MiR-145 is involved in the p53 signalling pathway. Downregulation of miR-145-5p oncogenes like FSCN1 (fascin actin-bundling protein 1) at the protein level or SOX2 causes migration, invasion, tumour growth, and EMT. Its minimal expression is correlated with advanced disease stages and poor survival rates [80]. Tumour-suppressive miRNAs can serve as biomarkers for early detection, prognosis, and monitoring the response to HNC therapy. Tumour growth and progression could be inhibited using miRNA or gene therapy approaches. Combining miRNA-based therapies with conventional chemotherapy and radiotherapy may change therapeutic outcomes and overcome resistances [100]. MicroRNAs’ unique expression profiles and functional roles make them valuable as diagnostic biomarkers and therapeutic targets for HNC. Further research into these miRNAs’ specific mechanisms and interactions in HNC will be essential for developing effective miRNA-based therapies and improving patient outcomes (Figure 2).

### 5.3. In Silico Validation of miRNAs in Head and Neck Cancer

To validate miRNAs, the OncoMir database (https://oncomir.org/) was utilized. This validation revealed that among the oncomiRs, hsa-miR-21-5p, hsa-miR-31-5p, hsa-miR-221-3p, hsa-miR-222-3p, hsa-miR-196a-5p, and hsa-miR-200c-3p are linked to tumourigenesis in HNSC (Table 5). For tumour-suppressive miRNAs, hsa-miR-375 and hsa-miR-145-5p were also associated with HNSC tumourigenesis (Table 6). Notably, hsa-miR-155-3p was the only miRNA significantly related to survival outcomes in HNSC (Table 7). Additionally, a strong correlation was observed between hsa-miR-155-3p and the genes RAI14, S1PR5, OSBPL10, and METTL6 (Table 7). 

## 6. Recent Studies Profiling miRNA Expression in Head and Neck Cancer Using High-Throughput Technologies

Several recent studies have applied high-throughput technologies to profile miRNA expression in HNC. These technologies are advanced and can be used to perform comprehensive and large-scale analyses of miRNA molecules. Using these technologies, the roles of miRNAs in cancer development and progression and their potential therapeutic applications have been revealed. Some of the key high-throughput technologies for miRNA profiling are next-generation sequencing (NGS), RNA-Seq (total RNA sequencing), microarrays, Quantitative Real-Time PCR (qRT-PCR), Digital PCR (dPCR), NanoString nCounter Analysis, and microfluidics-based platforms. The following are selected examples of recent achievements in finding key miRNAs in HNC using the aforementioned methods. In 2017, Ganci et al. identified 35 miRNAs that were deregulated in both tumour and peri-tumoural tissues in head and neck squamous cell carcinoma patients using microarray hybridization, reverse transcription of miRNAs, and qPCR. This identification of a predictive miRNA signature would be crucial for early detection, follow up, and treatment strategies for head and neck squamous cell carcinoma patients [101]. In 2023, three miRNA panels were identified with the potential for use in the diagnosis of HNSCC from the patients’ plasma. This research indicated that miR-95-3p and miR-579-5p expression is significantly upregulated, and miR-1298-3p expression is downregulated in the plasma of HNSCC patients. This investigation shows a convenient diagnostic and screening tool for the early detection of head and neck cancer [102]. Another method is dPCR which assists in the management of patients with a cancer of unknown primary site (CUP). Laprovitera et al. predicted the primary site of 53 CUPs using a molecular assay based on digital miRNA profiling. Furthermore, HNC with metastases of known origin was assessed in this study and the unknown sites were identified in the CUP patients using dPCR. High-throughput technologies have revolutionized miRNA profiling in HNC. These methods provide comprehensive, sensitive, and quantitative data that are essential for understanding the roles of miRNAs in cancer and developing diagnostic and therapeutic strategies [103]. Each technology has its unique advantages, and the choice of method depends on the specific research question, available resources, and the desired resolution of the miRNA profiling. These unique methodologies have identified numerous miRNAs that are expressed in various HNC subtypes, providing sensitive and precise miRNA profiling to improve HNC diagnostic and therapeutic approaches. 

Despite all these advancements, HNC is still diagnosed at advanced stages in many cases, generating limited improvements in survival rates. The improvement of miRNA therapeutics in HNC faces several limitations such as contradictions in study results, miRNA specificity issues, a limited understanding of the tumour microenvironment (TME), heterogeneity in molecular subtypes of HNC, a lack of effective biomarkers for early detection, resistance of miRNAs to radio- and chemo-therapies, differences between HPV-positive vs. HPV-negative HNC, the detection and prognostic challenges of miRNAs, and epidemiological study limitations [104]. These limitations highlight the need for further research to improve the specificity and reliability of miRNA-based diagnostics and therapeutics in HNC. However, there are several promising fields that the future of profiling miRNAs in HNC can focus on. Investigating the miRNA profiles specific to different anatomical sites of HNC (miRNA signatures) is a promising approach that can help identify unique biomarkers and therapeutic targets [105]. In addition, detecting miRNA through liquid biopsies, such as saliva or blood samples, is a non-invasive diagnostic and prognostic tool for HNC, potentially allowing for early detection and the monitoring of treatment response in HNC patients [106,107]. The optimization of delivery mechanisms and the stability of miRNA therapeutics are paramount to allow for the wide application of miRNAs in HNC [107]. Moreover, integrating miRNA profiling with comprehensive genomic and transcriptomic data will provide insights into the molecular mechanisms driving HNC. Ultimately, immunotherapy is a very promising approach for the future of HNC treatment. Studying how miRNAs interact with the immune system in HNC and identifying miRNAs that enhance the immune response or can serve as biomarkers for immunotherapy are fundamental in shaping the future of miRNA therapeutic research in HNC [106].

In summary, research into miRNA profiling in head and neck cancer offers exciting opportunities for improving HNC diagnosis and treatments. By focusing on specific miRNA profiles, integrating genomic data, and exploring non-invasive diagnostic methods, uncovering new insights and therapeutic targets will be possible. All these efforts will ultimately serve to enhance patient care and outcomes.

## 7. Future Perspectives

Precision medicine in HNC detection and treatment focuses on tailoring treatment to individual genetic profiles. In HNC, identifying specific genetic mutations and viral associations, particularly HPV, has led to the development of targeted therapies and vaccines designed to improve patient outcomes. For example, the presence of HPV in oropharyngeal cancer has been associated with a better prognosis and response to certain therapies. Targeted treatments, such as those directed against the epidermal growth factor receptor (EGFR), and the use of HPV vaccines are key advances in the precision medicine approach to HNC [10].

The future of precision medicine lies in further integrating genomic, transcriptomic, metabolomic, and proteomic data to enable personalized treatment approaches [108,109]. Combining these different data types allows clinicians to develop more effective, individualized treatment plans that improve patient outcomes and reduce adverse effects. In HNC, the research has focused on targeting EGFR with monoclonal antibodies such as cetuximab and exploring the therapeutic potential of inhibiting the PI3K/AKT/mTOR signalling pathway. These signalling pathways are critical for regulating cell growth, survival, and proliferation, and their dysregulation is a hallmark of many cancers. In addition, immunotherapy based on checkpoint inhibitors, such as pembrolizumab and nivolumab, is increasingly recognized as a potential treatment strategy in HNC. These therapies use the patient’s immune system to recognize and destroy cancer cells, offering a new avenue for treatment, especially in cases where conventional therapies have failed [14,110].

Laboratory studies and computer-based data analyses are at the forefront of revolutionizing cancer research and the treatment of HNC. The use of artificial intelligence (AI) and machine learning (ML) analysis of large data sets can potentially discover new biomarkers, predict treatment outcomes, and personalize treatment plans. These technologies can process and interpret vast amounts of genomic and clinical data, identifying patterns and correlations that may not be apparent using traditional methods. Big data analytics enable the integration of different data types and provide comprehensive insights into cancer biology and treatment responses [111]. In addition, developing digital health tools and wearable technologies for real-time patient monitoring offers new opportunities to improve quality of life and treatment adherence. These tools can track vital signs, medication adherence, and symptom progression, enabling timely interventions and adjustments to treatment plans. Integrating these technologies into clinical practice is expected to accelerate the pace of discoveries in cancer research and change the perspective of cancer care so that treatments become more personalized and responsive to individual patient needs [112].

The future of HNC research and treatment is poised for significant advances through precision medicine, new therapeutic targets, and the integration of technology and data science. These developments promise to improve patient outcomes, personalize cancer treatment, and ultimately change the approach to treating these complex diseases. The continued evolution of precision medicine, driven by genomic, transcriptomic, metabolomic, and proteomic insights, will likely lead to more targeted and effective treatments. The discovery and validation of novel therapeutic targets will expand the arsenal of available treatments. At the same time, advances in AI, ML, and digital health technologies will improve the precision and personalization of cancer care. Integrating these innovative approaches will be instrumental in minimizing overdiagnosis and overtreatment, improving survival rates, and enhancing the quality of life for patients with HNC. Collaboration between researchers, clinicians, and technicians will be crucial in realizing the full potential of these advancements and ensuring that each patient receives the most effective and personalized therapy.

## 8. Conclusions

This review highlights the significant advances in understanding the molecular complexity of head and neck cancer (HNC), with a particular focus on microRNAs (miRNAs). miRNAs have been shown to be crucial regulators of gene expression and are promising biomarkers for diagnosis, prognosis, and therapeutic intervention in HNC. Their central role in the molecular landscape of these malignancies underscores their potential for use in personalized treatment strategies that can improve patient outcomes while minimizing the toxicity associated with conventional therapies. The review also highlighted the potential of miRNAs in improving the precision of the diagnosis and treatment of HNC. Advances in miRNA profiling have not only improved the accuracy of diagnosis of HNC, but have also facilitated the identification of new therapeutic targets. These developments are key to the evolving field of precision medicine in which treatment strategies are tailored to the individual molecular characteristics of each patient’s tumour. The transition from traditional, blanket approaches to more patient-specific therapies represents an important milestone in cancer treatment and offers hope for improved survival rates and a better quality of life for patients.

Despite the promising advances in miRNA research, challenges remain, including conflicting results in the literature. These inconsistencies are often due to small sample sizes, different detection techniques, and a lack of standardization between studies. For example, miR-21 and miR-155 have been identified as oncogenic in multiple studies, but the differences in reported expression levels and clinical relevance underscore the need for large-scale, multi-centre validation studies. Addressing these limitations through standardized methods and collaborative research efforts will be critical to ensuring that miRNAs can be reliably integrated into clinical practice. The molecular understanding of HNC, particularly through the study of miRNAs, has contributed significantly to the development of new therapeutic strategies. By exploring the genetic and epigenetic changes regulated by miRNAs, researchers have identified important signalling pathways and targets for intervention. These findings are crucial for developing more effective treatments and overcoming resistance to current therapies. The more miRNAs are explored, the more important their role will become in designing personalized treatment plans that will drive the next wave of advances in cancer treatment. Looking ahead, continued research and collaboration within the oncology community is critical. Multidisciplinary efforts that combine the expertise of molecular biologists, clinicians, data scientists, and patient advocates will accelerate the discovery of new biomarkers and therapeutic targets. Such collaborations will also ensure that breakthroughs in miRNA research are rapidly translated into clinical practice to improve patient care. These collaborative efforts are key to overcoming the remaining challenges in the treatment of HNC, including drug resistance, metastasis, and relapse.

Although significant progress has been made in understanding miRNAs and their role in HNC, further research, innovation, and collaboration are needed to fully unlock their potential. As we continue to explore the molecular basis of HNC through the lens of miRNAs, the future of cancer treatment will increasingly focus on precision medicine. This transformation will be driven by the integration of cutting-edge technologies, data science, and personalized approaches to ensure that each patient receives the most effective and targeted treatment.

## Figures and Tables

**Figure 1 cancers-16-03716-f001:**
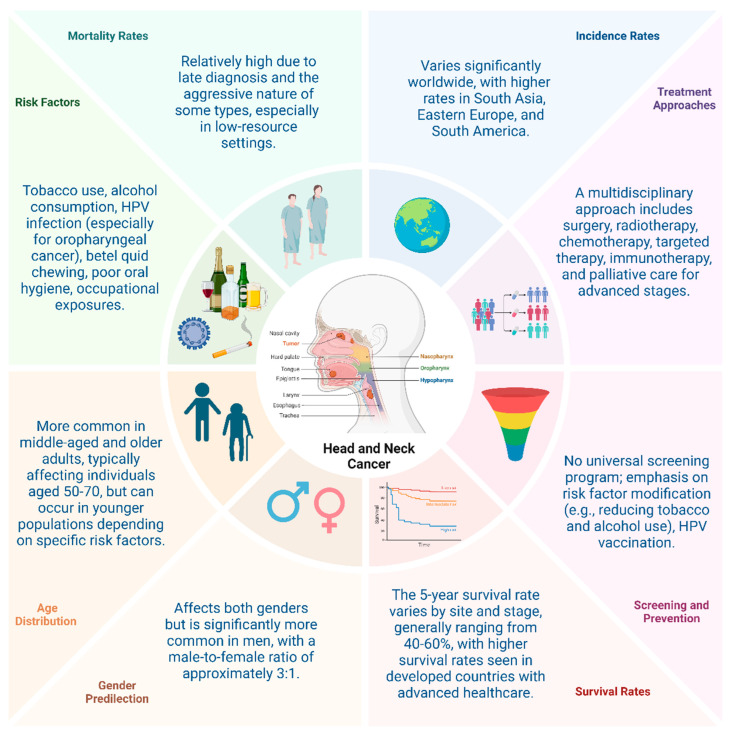
Epidemiological aspects of head and neck cancer (created with BioRender.com).

**Figure 2 cancers-16-03716-f002:**
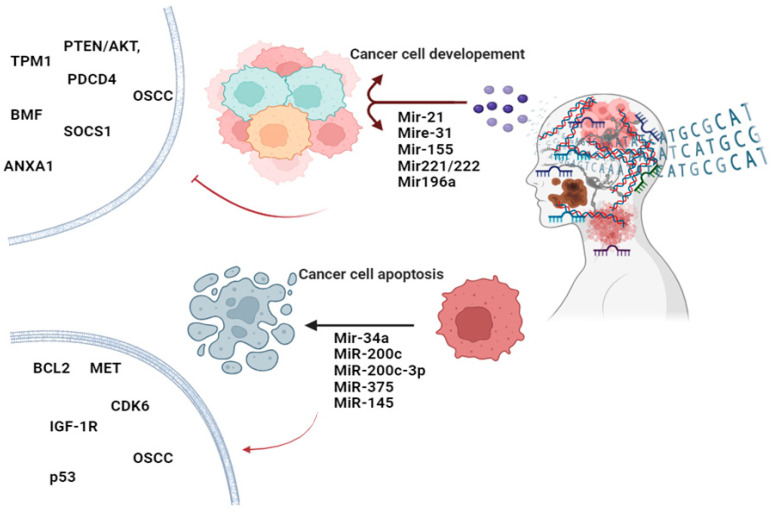
OncomiRs and tumour-suppressive miRNAs: key regulators in head and neck cancer progression. This figure illustrates the dual roles of oncomiRs and tumour-suppressive miRNAs in head and neck cancer. OncomiRs promote tumour growth and metastasis, while tumour-suppressive miRNAs inhibit cancer progression. Together, these miRNAs orchestrate the molecular pathways in cell proliferation, apoptosis, and angiogenesis (Created with BioRender.com).

**Table 1 cancers-16-03716-t001:** Molecular subtypes of head and neck cancer: characteristics and clinical implications.

Molecular Subtype	Characteristic Features	Clinical Implications	Reference
HPV-positive	Presence of HPV DNA, overexpression of p16 (a surrogate marker for HPV oncogenic activity), and absence of TP53 mutations. Typically associated with oropharyngeal cancers.	HPV-positive tumours respond better to radiation and chemotherapy, and patients have a better overall prognosis. These tumours also have a reduced risk of recurrence compared to HPV-negative tumours.	[20]
Basal	High expression of basal cell markers, frequent EGFR amplification, and TP53 mutations. These tumours often show poor differentiation.	Poor prognosis due to aggressive tumour behaviour and resistance to certain therapies. EGFR inhibitors, like cetuximab, may be used, but overall survival remains low.	[21]
Mesenchymal	Characterized by epithelial-to-mesenchymal transition (EMT), with expression of mesenchymal markers such as vimentin and N-cadherin, and reduced E-cadherin levels.	High resistance to conventional treatments like chemotherapy and radiation therapy. This subtype is typically more invasive and associated with metastatic potential.	[22]
Atypical	Defined by mutations in NOTCH1 and low expression of HPV and p16. Exhibits molecular heterogeneity.	Variable prognosis; potential sensitivity to NOTCH inhibitors in the future. These tumours may benefit from targeted therapies, but their response to current treatments is unpredictable.	[23]
Classical	High expression of genes involved in cell cycle regulation, including cyclin D1 and CDK6, with frequent alterations in TP53.	Highly aggressive with rapid tumour progression. Targeting cell cycle pathways with CDK inhibitors has shown potential for treatment in clinical trials.	[24]
Immune-Related	High levels of immune cell infiltration, particularly lymphocytes, and expression of immune checkpoint molecules like PD-L1.	These tumours show potential responsiveness to immunotherapy, particularly immune checkpoint inhibitors (e.g., PD-1/PD-L1 inhibitors like pembrolizumab or nivolumab). Immunotherapy has shown promise in improving survival rates.	[25]
Hypoxic	Overexpression of hypoxia-inducible factors (HIFs), leading to an adaptive response to low oxygen environments. These tumours are typically resistant to apoptosis and have enhanced angiogenesis.	Poor prognosis due to treatment resistance and aggressive progression. Targeting hypoxic pathways and angiogenesis inhibitors may offer therapeutic benefits.	[26]
Metabolic	Characterized by alterations in metabolic pathways, including increased glycolysis (Warburg effect) and overexpression of glucose transporter 1 (GLUT1).	Potential for therapies targeting metabolic pathways (e.g., inhibitors of glycolysis or metabolic enzymes). These tumours may also be resistant to conventional therapies, making metabolic inhibitors a promising approach.	[27]

**Table 2 cancers-16-03716-t002:** Key proteins and genes in head and neck cancer (HNC).

Key Protein/Gene	Role in Cancer	Clinical Relevance	Ref.
HPV E6/E7	HPV E6 inactivates p53 and HPV E7 inactivates Rb, leading to uncontrolled cell cycle progression and tumourigenesis.	HPV status is a critical prognostic marker and determines treatment strategies.	[33]
EGFR	Overexpressed in many HNCs, leading to increased cell proliferation.	Targeted by EGFR inhibitors like cetuximab, it predicts responsiveness to therapy.	[34]
TP53	Mutations lead to loss of tumour suppressor function, contributing to carcinogenesis.	Associated with poor prognosis and aggressive disease; potential target for therapy.	[35]
CDKN2A (p16)	Tumour suppressor gene, loss contributes to cell cycle deregulation.	Frequently mutated or deleted in HNC, indicative of poor prognosis.	[36]
PIK3CA	Mutation activates the PI3K/AKT pathway, promoting tumourigenesis.	Target for PI3K inhibitors associated with therapeutic resistance.	[37]
TIMPs (Tissue Inhibitors of Metalloproteinases)	Regulate ECM remodelling and metastasis by inhibiting MMPs, potentially suppressing tumour progression.	Targets for therapy and markers for disease progression and response to treatment.	[38]
Gal-3 (Galectin-3)	Involved in cell adhesion, migration, and tumour progression.	Potential marker for prognosis and therapeutic targeting.	[39]
MMPs (Matrix Metalloproteinases)	Facilitate tumour invasion and metastasis through ECM degradation.	Biomarkers for invasive potential and therapeutic targets.	[40]
Fibronectin	Contributes to cell adhesion and migration, influencing tumour growth and metastasis.	Insights into tumour progression and potential therapeutic implications.	[41]

**Table 3 cancers-16-03716-t003:** Characterization of top DEGs identified for HNC.

Gene	Role in Cancer	References
*AURKA*	High expression is associated with cancer progression, dysregulated proliferation, and inhibition of apoptosis. It is a good predictor of OS, with overexpression representing a poor prognosis.	[42,43,44]
*HMGA2*	Overexpression increases tumour cell proliferation, migration, invasion, and metastasis. A good predictor of OS, with high levels indicating poor prognosis.	[45,46,47]
*MMP1*	Upregulation is associated with ECM degradation and consequently promotes cancer invasiveness. Its expression correlates with T stage (TNM classification) and cancer progression.	[48,49]
*PLAU*	Overexpression is associated with ECM degradation, an increased risk of developing HNC, and an upregulation of MMP1. Knockdown inhibits proliferation, invasion, and metastasis.	[49,50]
*SERPINE1*	Overexpression is associated with N stage (TNM classification), poor DFS, and the promotion of radiotherapy resistance. Inhibition leads to decreased cell proliferation and invasion.	[4,51,52]

**Table 4 cancers-16-03716-t004:** Diagnostic and prognostic biomarkers in HNC: effectiveness overview.

Biomarker	Diagnostic Use	Prognostic Use	Potential Effectiveness	Ref.
HPV DNA	Identifies HPV-associated HNC	Indicates better prognosis and response to treatment in HPV-positive cases	Highly effective for subclassification and prognosis	[72]
EGFR	Used for identifying tumours with EGFR overexpression	Associated with poor response to radiation and certain chemotherapies	Effective in selecting candidates for EGFR-targeted therapies	[73]
p16	Surrogate marker for HPV oncogenic activity	Suggests improved outcomes in HPV-positive HNC	Widely used, offers good prognostic value	[74]

**Table 5 cancers-16-03716-t005:** miRNAs associated with tumourigenesis of HNSC.

miRNA Name	*p*-Value *t*-Test	FDR *t*-Test	Upregulated in	Tumour Log2 Mean Expression	Normal Log2 Mean Expression
hsa-miR-21-5p	8.64 × 10^−15^	6.65 × 10^−13^	Tumour	0	0
hsa-miR-31-5p	1.12 × 10^−7^	8.23 × 10^−7^	Tumour	0	0
hsa-miR-221-3p	1.14 × 10^−3^	3.09 × 10^−3^	Tumour	0	0
hsa-miR-222-3p	2.07 × 10^−6^	1.02 × 10^−5^	Tumour	0	0
hsa-miR-196a-5p	2.44 × 10^−17^	8.45 × 10^−15^	Tumour	0	0
hsa-miR-200c-3p	1.44 × 10^−2^	3.12 × 10^−2^	Tumour	0	0
hsa-miR-375	1.77 × 10^−11^	3.60 × 10^−10^	Normal	0	0
hsa-miR-145-5p	2.44 × 10^−4^	7.54 × 10^−4^	Normal	0	0

**Table 6 cancers-16-03716-t006:** miRNAs significantly associated with survival in HNSCC.

miRNA Name	Log Rank *p*-Value	Log Rank FDR	Z-Score	Upregulated in	Deceased Log2 Mean Expression	Living Log2 Mean Expression	*t*-Test *p*-value	*t*-Test FDR
hsa-miR-155-3p	3.46 × 10^−2^	3.77 × 10^−1^	1.985	Surviving patients	0.52	0.65	3.33 × 10^−1^	5.94 × 10^−1^

**Table 7 cancers-16-03716-t007:** hsa-miR-155-3p–target correlations in HNSCC.

Gene	Gene Description	Correlation	Correlation *p*-Value	Correlation FDR	miRDB Score
RAI14	Retinoic acid-induced 14	−0.106	1.73 × 10^−2^	5.67 × 10^−1^	82
S1PR5	Sphingosine-1-phosphate receptor 5	−0.0983	2.73 × 10^−2^	5.67 × 10^−1^	76
OSBPL10	Oxysterol binding protein-like 10	−0.0961	3.09 × 10^−2^	5.67 × 10^−1^	90
METTL6	Methyltransferase-like 6	−0.0876	4.93 × 10^−2^	5.79 × 10^−1^	63

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
