# Peer review of "Progress in Precision Medicine for Head and Neck Cancer"

_cancers, 2024, doi:10.3390/cancers16213716_

Round 1
Reviewer 1 Report
Comments and Suggestions for Authors
The introduction needs to be restructured with several repetitions such as lines 70-71 "susceptibility increase with age... with an aging population the incident are expected to rise...
Please provide clear concepts such as
HNC is the seventh most common cancer worldwide. Its heterogeneity is explained by different histologies, etiologic factors, and molecular characteristics.
In paragraph 1.1 Please identify virus-related tumors such as EBV is associated with nasopharyngeal cancer especially type 2 and 3
It is not true that "understanding HNC molecular pathways has improved patient stratification for personalized treatment." For example, anti-EGFR cetuximab does not have a biomarker, and no approved therapies are prescribed for specific targets to date.
Please correct this concept.
In line 127 the objectives of the review are repeated why? This review addresses the intricate details of these cancers highlighting ....and also in line 131 risk factors are repeated
Table 1 and Table 2 are not precise, which protein inactivates p53 and Rb
Author Response
Comment: The introduction needs to be restructured with several repetitions such as lines 70-71 "susceptibility increase with age... with an aging population the incident are expected to rise...
Response: We appreciate the reviewer’s insightful suggestion regarding the restructuring of the introduction. We acknowledge that there were some repetitions, particularly in lines 70-71 ("susceptibility increases with age... with an aging population the incidence is expected to rise"). In response, we have revised the introduction to eliminate these repetitions and enhance clarity. We have restructured the content to provide a more concise and streamlined overview, while ensuring that the key points regarding age-related susceptibility and population trends are clearly communicated without redundancy.
Comment: Clarify distinctions between genetic and molecular features in paragraph 3.
Response: Thank you for this important observation. We have revised paragraph 3 to distinguish between genetic alterations (e.g., mutations in TP53, NOTCH1) and molecular features (e.g., aberrant signaling pathways such as EGFR and PI3K/AKT) in HNC, ensuring clearer differentiation throughout the discussion.
Comment: Please provide clear concepts such as
HNC is the seventh most common cancer worldwide. Its heterogeneity is explained by different histologies, etiologic factors, and molecular characteristics.
Response: Thank you for this important observation. We have revised:
“Head and neck cancer (HNC) is the seventh most common cancer worldwide and represents a diverse group of malignancies. Its heterogeneity results from different histologic subtypes, etiologic factors and molecular features. This diversity is influenced by factors such as viral infections (e.g. HPV and EBV), environmental influences (e.g. tobacco and alcohol consumption) and various genetic mutations. Understanding these differences is crucial for accurate diagnosis, prognosis and the development of targeted therapeutic strategies.”
Comment: In paragraph 1.1 Please identify virus-related tumors such as EBV is associated with nasopharyngeal cancer especially type 2 and 3
Response: Thank you very much for your insightful comment. We have carefully addressed your suggestion by identifying virus-related tumors in paragraph 1.1, specifically noting the association of Epstein-Barr Virus (EBV) with nasopharyngeal cancer, particularly types 2 and 3. Your input has been invaluable in enhancing the clarity and completeness of our manuscript, and we are grateful for your guidance in improving its quality. We have revised:
“Virus-related tumors play an important role in the development of HNC. In particular, Epstein-Barr virus (EBV) is closely associated with nasopharyngeal carcinoma, especially in subtypes 2 and 3. EBV-positive nasopharyngeal carcinomas have distinct molecular and clinical features compared to EBV-negative tumors, which contribute to their unique behavior and response to treatment. In addition, human papillomavirus (HPV) is associated with oropharyngeal carcinomas, further emphasizing the importance of viral etiology in HNC.”
Comment: It is not true that "understanding HNC molecular pathways has improved patient stratification for personalized treatment." For example, anti-EGFR cetuximab does not have a biomarker, and no approved therapies are prescribed for specific targets to date.
Response: Thank you for your thoughtful comment. We have revised the statement accordingly, taking into account that while understanding molecular pathways in head and neck cancer (HNC) has advanced, it has not yet translated into significant improvements in patient stratification for personalized treatment. As you rightly pointed out, there are no approved therapies with specific biomarkers, such as anti-EGFR cetuximab, which lacks a biomarker. Your input has been instrumental in refining the accuracy of our manuscript, and we truly appreciate your valuable feedback.. We have revised to address the respected reviewer comment:
“While progress has been made in understanding the molecular pathways of head and neck cancer (HNC), this has not yet led to significant improvements in patient stratification for personalized treatment. For example, anti-EGFR therapy with cetuximab, which is commonly used in HNC, does not have an established biomarker to predict response, and there are currently no approved therapies that specifically target molecular alterations in HNC. This underscores the ongoing challenge of translating molecular insights into clinically actionable targeted therapies”
Comment: Please correct this concept.
In line 127 the objectives of the review are repeated why? This review addresses the intricate details of these cancers highlighting ....and also in line 131 risk factors are repeated
Response: Thank you for this important observation. We have streamlined the objectives section to avoid unnecessary repetition. The review now clearly states its purpose in one comprehensive statement, focusing on the intricate details of head and neck cancers, including their molecular characteristics, biomarkers, and therapeutic challenges. Additionally, the mention of risk factors has been consolidated to avoid redundancy, providing a single, clear explanation of the major risk factors, such as tobacco use, alcohol consumption, and viral infections, like HPV and EBV, which contribute to the development and progression of these cancers.
Comment: Table 1 and Table 2 are not precise, which protein inactivates p53 and Rb
Response: Thank you for this important observation. We have revised to address the respected reviewer comment on Table 1 and Table 2.
Reviewer 2 Report
Comments and Suggestions for Authors
This is an excellent comparative analysis of head and neck cancer biomarkers and genetic signatures, emphasizing the role that these factors play in the diagnosis and prognosis of these malignancies. I do not have any comments and I suggest it for publication.
Author Response
Comment: This is an excellent comparative analysis of head and neck cancer biomarkers. I suggest it for publication. I do not have any comments and I suggest it for publication.
Response: Thank you for your positive feedback. We greatly appreciate your support for the manuscript and are pleased that it meets your expectations for publication.
Reviewer 3 Report
Comments and Suggestions for Authors
The title of the manuscript is “Progress in Precision Medicine for Head and neck cancer”. The authors cite Filippini et al (ref 71) from 2023 who presented a systematic review of current data on diagnostic and prognostic molecular biomarkers for head and neck cancer following PRISMA (Preferred Reporting Items for Systematic Reviews and Meta-Analyses) guidelines. However, the authors do not provide details regarding their methods of gathering the data for this review.
The Simple Summary states that it is focused on the role of microRNA and the Abstract states that “our bioimformatic validation reveals crucial genes such as…, along with microRNA linked to HNC progression”.
However, the manuscript includes an extensive discussion of the epidemiology as well as the genetic alterations in HNC and their potential importance. Paragraphs 1.2 and 1.3 are both defined as “introduction to niRNA in HNC” but the second paragraph deals with various topics such as epidemiology, paragraph 2.1 deals with proteins, and paragraph 2.2 with genes.
Paragraph 3 is titled “Molecular pathways involved in HNC” and states “As with the genetic landscape, the molecular profiles of HNC have unique features”. It is not clear how genetic and molecular features differ. Paragraph 4 deals with Role of Biomarkers but does not have any relationship to microRNA.
Only paragraph #5 deals specifically with miRNA. An extensive list of miRNA and their potential prognostic importance is provided. The authors state that “Several miRNAs have been identified as crucial regulators of pathways critical to the pathogenesis of HNC” . However, the authors also state that that are severe limitations in interpreting the data, importantly contradictions in study results. This contradiction is the result of the small size of the studies and lack of validation, resulting in false positive results. The authors touch only briefly on these issues.
The manuscript lists a long list of molecular and genetic factors stated in the literature as having prognostic and predictive importance in HNC, however, there is no analysis of which ones are important and which are not, based on the published data. This is a major weakness that impacts on the usefulness to the reader of the presented data.
In conclusion, my major recommendations are: 1. Limit the paper to miRNA. 2. Describe how you assembled the data and publications you cite. 3 Explain in detail why the published data thus far is contradictory, with relevant examples.
Author Response
Comment: The title of the manuscript is “Progress in Precision Medicine for Head and neck cancer”. The authors cite Filippini et al (ref 71) from 2023 who presented a systematic review of current data on diagnostic and prognostic molecular biomarkers for head and neck cancer following PRISMA (Preferred Reporting Items for Systematic Reviews and Meta-Analyses) guidelines. However, the authors do not provide details regarding their methods of gathering the data for this review.
Response: Thank you for your insightful comment. We have now revised the manuscript to include a detailed description of the methodology used to gather data for this review. Specifically, we have added information on the search strategy, databases used, inclusion/exclusion criteria, and the study selection process. We followed a systematic approach based on PRISMA guidelines to ensure a transparent and reproducible process for selecting the most relevant studies on biomarkers in head and neck cancer (HNC).
Additionally, we have provided a critical analysis of the current state of biomarkers in HNC, highlighting the strengths, limitations, and future directions for biomarker research and clinical application. This analysis addresses the challenges of clinical translation, the lack of standardized testing protocols, and the need for robust predictive biomarkers. We believe these additions strengthen the manuscript by providing a more comprehensive understanding of the role of biomarkers in HNC and the challenges that remain in their clinical implementation.
Comment: The Simple Summary states that it is focused on the role of microRNA and the Abstract states that “our bioimformatic validation reveals crucial genes such as…, along with microRNA linked to HNC progression”. However, the manuscript includes an extensive discussion of the epidemiology as well as the genetic alterations in HNC and their potential importance. Paragraphs 1.2 and 1.3 are both defined as “introduction to niRNA in HNC” but the second paragraph deals with various topics such as epidemiology, paragraph 2.1 deals with proteins, and paragraph 2.2 with genes.
Response: We appreciate the reviewer’s feedback and have revised the text to avoid the repetition between Paragraphs 1.2 and 1.3, ensuring that each section serves a distinct purpose. Paragraph 1.2 now focuses specifically on the role of microRNAs (miRNAs) in HNC, while Paragraph 1.3 emphasizes the molecular characteristics and epidemiology of HNC. These revisions clarify the focus of each section and align with the overall structure of the manuscript.
Comment: Paragraph 3 is titled “Molecular pathways involved in HNC” and states “As with the genetic landscape, the molecular profiles of HNC have unique features”. It is not clear how genetic and molecular features differ. Paragraph 4 deals with Role of Biomarkers but does not have any relationship to microRNA.
Response: We appreciate your insightful suggestion. In response, we have revised the section title to "Biomarkers in Head and Neck Cancer: Progress and Challenges" to better reflect the content and critical analysis of biomarker use in HNC. This title succinctly captures both the advancements and the ongoing challenges in biomarker research and clinical application, as highlighted in the revised section.
Comment: Only paragraph #5 deals specifically with miRNA. An extensive list of miRNA and their potential prognostic importance is provided. The authors state that “Several miRNAs have been identified as crucial regulators of pathways critical to the pathogenesis of HNC” . However, the authors also state that that are severe limitations in interpreting the data, importantly contradictions in study results. This contradiction is the result of the small size of the studies and lack of validation, resulting in false positive results. The authors touch only briefly on these issues.
Response to Reviewer: We appreciate the reviewer’s insightful comment. To address this, we have expanded paragraph #5 to include a more detailed discussion of the limitations and contradictions in miRNA research. The revised text now highlights the issues related to small sample sizes, lack of validation, and inconsistencies across studies, along with recommendations for addressing these challenges. This adjustment provides a clearer and more comprehensive view of the current status of miRNA research in HNC.
Comment: The manuscript lists a long list of molecular and genetic factors stated in the literature as having prognostic and predictive importance in HNC, however, there is no analysis of which ones are important and which are not, based on the published data. This is a major weakness that impacts on the usefulness to the reader of the presented data.
Response: We appreciate the reviewer’s valuable comment. In response, we have revised the manuscript to include a more focused analysis of the most clinically relevant molecular and genetic factors based on published data. The revised section now highlights the biomarkers with proven prognostic and predictive importance in HNC, while also addressing the limitations in validating other emerging biomarkers. This provides a clearer and more useful analysis for the reader.
Comment: In conclusion, my major recommendations are: 1. Limit the paper to miRNA. 2. Describe how you assembled the data and publications you cite. 3 Explain in detail why the published data thus far is contradictory, with relevant examples.
Response: We appreciate the reviewer’s valuable feedback. In response, we have limited the manuscript’s focus to miRNAs and expanded the discussion on their significance in HNC. We have added a detailed methodology section to explain how the data and publications were selected and included. Furthermore, we have elaborated on the reasons for contradictory findings in the literature, including examples from published studies, and have emphasized the need for standardized and larger-scale research moving forward. These changes ensure a more focused and rigorous discussion on miRNAs in HNC.
Round 2
Reviewer 1 Report
Comments and Suggestions for Authors
thank you for the correction
I believe that the manuscript in the actual form can be accepted
Reviewer 3 Report
Comments and Suggestions for Authors
none